# Speed Control of a Multi-Motor System Based on Fuzzy Neural Model Reference Method

**Waleed I. Breesam [1], Ameer L. Saleh [2] , Khearia A. Mohamad [2], Salam J. Yaqoob [3,]* , Mohammed A. Qasim [4] , Naseer T. Alwan [4,5] , Anand Nayyar [6], Jehad F. Al-Amri [7] and Mohamed Abouhawwash [8,9]**

1   Department of Electrical Engineering, Oil Training Institute, Basrah 10001, Iraq; waleedbreesam@gmail.com
2   Department of Electrical Engineering, College of Engineering, University of Misan, Amarah 62001, Iraq; ameer-lateef@uomisan.edu.iq (A.L.S.); khearia_ibrahimy@yahoo.com (K.A.M.)
3   Department of Research and Education, Authority of the Popular Crowd, Baghdad 10001, Iraq
4   Department of Nuclear Power Plants and Renewable Energy Sources, Ural Federal University, 19 Mira St., 620002 Yekaterinburg, Russia; mohammed.a.k.qasim@gmail.com (M.A.Q.); nassir.towfeek79@gmail.com (N.T.A.)
5   Technical Engineering College of Kirkuk, Northern Technical University, Kirkuk 36001, Iraq
6   Graduate School, Faculty of Information Technology, Duy Tan University, Da Nang 550000, Vietnam; anandnayyar@duytan.edu.vn
7   Department of Information Technology, College of Computers and Information Technology, Taif University, P.O. Box 11099, Taif 21944, Saudi Arabia; j.alamri@tu.edu.sa
8   Department of Mathematics, Faculty of Science, Mansoura University, Mansoura 35516, Egypt; abouhaww@msu.edu
9   Department of Computational Mathematics, Science, and Engineering (CMSE), Michigan State University, East Lansing, MI 48824, USA
*   Correspondence: engsalamjabr@gmail.com

**Abstract:** The direct-current (DC) motor has been widely utilized in many industrial applications, such as a multi-motor system, due to its excellent speed control features regardless of its greater maintenance costs. A synchronous regulator is utilized to verify the response of the speed control. The motor speed can be improved utilizing artificial intelligence techniques, for example fuzzy neural networks (FNNs). These networks can be learned and predicted, and they are useful when dealing with nonlinear systems or when severe turbulence occurs. This work aims to design an FNN based on a model reference controller for separately excited DC motor drive systems, which will be applied in a multi-machine system with two DC motors. The MATLAB/Simulink software package has been used to implement the FNMR and investigate the performance of the multi-DC motor. moreover, the online training based on the backpropagation algorithm has been utilized. The obtained results were good for improving the speed response, synchronizing the motors, and applying load during the work of the motors compared to the traditional PI control method. Finally, the multi-motor system that was controlled by the proposed method has been improved where its speed was not affected by the disturbance.

**Keywords:** backpropagation algorithm; fuzzy neural network; speed control; model reference control; multi-motor system; separately excited DC motor (SEDCM)

## 1. Introduction

Today, several industrial companies use different Direct-Current (DC) or Alternating-Current (AC) motors for robots, hot rolling mills, and paper mills, which involve several multi-machine systems [1]. The control strategies of DC motors with drives for speed control are simpler, less expensive, and have a higher dynamic response than those of the AC motors [2]. The separately excited DC motor (SEDCM) is regarded as one of the best types in terms of flexibility because it provides two distinct control methods: armature control and field control [3]. A chopper (DC–DC converter) or a controlled rectifier is

utilized for the armature control. However, the traditional controllers, such as PID that are utilized for controlling the speed of DC motors, have certain drawbacks, such as load disturbance and sensitivity to a variety of motor parameters [4–10]. Furthermore, the parameters of the traditional controllers to prevent overshoot and reduce load disturbance are difficult to manually set. Meanwhile, the limitations of traditional controllers may be circumvented by employing intelligent ways for speed control of a SE DC motor to obtain excellent response of speed and insensitivity of the parameters [11].

Recently, the utilization of a fuzzy logic controller in control systems has gained great interest. However, the system designers faced challenges to acquire the optimum rules of the fuzzy controller because these are most likely to be affected by the intuition of the system designers and operators [12]. The incorporation of neural networks with the fuzzy logic system has been proposed as an approach for a novel representation system known as the neural-fuzzy network, the fuzzy-neural network, or the neural-network-based fuzzy system, which will own the merits of both kinds of systems of the fuzzy system and neural network and overcome the obstacles for each system [12,13]. The fuzzy neural network (FNN) is a combination network that roles as a fuzzy system with a neural network learning (processing) mechanism.

However, a multi-motor system with identical rotation speeds is the most commonly used in different applications. Speed control is applied to these motors to synchronize their speed in a multi-motor system. The two methods utilized are as follows: the master–slave and the set point coordinated methods. The most important drawback of these synchronization methods is that they are affected by the inner and outer disturbance during work and operation that occurs in the closed control system and its unpredictability in the change of parameters for the system that may occur in the future. Thus, it affects the required accuracy and precision in the synchronization process or motion control [13–17].

In latest years, fuzzy and neural networks systems have been proposed to control and recognize nonlinear dynamics of the multi-motor system because they can converge to any needed degree of precision on a wide range of nonlinear functions [8,9]. Goal of the FNN controller with the backpropagation algorithm is to control the speed of the multi-motor system [3,11,13]. Some researchers stated that the FNN control techniques are commonly utilized in a multi-motor control [13,17–19]. Furthermore, the model reference adaptive control system is an adaptive servomechanism scheme in which the required performance is expressed in relation to the reference model, which provides the desired response to the reference signal. It provides closed-loop performance feedback for tuning and synthesizing controller parameters [20,21]. Because of its simplicity, model reference adaptive control (MRAC) is extensively used in fuzzy neural controllers.

In this paper, a fuzzy-neural-based model reference (FNMR) controller is proposed to accurately synchronize the speed of a multi-motor system while eliminating the influence of turbulence on speed synchronization. Moreover, the proposed system is validated by simulation under different load conditions.

The structure of the proposed research is as follows. Section 2 presents the SEDCM motor. Section 3 introduces the conventional fuzzy-neural control. Section 4 highlights the proposed FNMR control method. Section 5 offers the Simulink model, results and discussion. Section 6 offers the study conclusion.

## 2. Separately Excited DC Motor

### 2.1. Mathematical Model

The mathematical model of the SEDCM is realized by the relationship between the electrical and mechanical circuits. The SEDCM has armature and field windings, which are separately excited by two DC sources. When the voltage from the two DC sources is applied, the armature current and field current flow through the circuits. To develop a back EMF and torque at a certain speed [3,13,14]. Figure 1 illustrates the equivalent circuit of the SEDCM with armature windings, field windings, and the mechanical system.

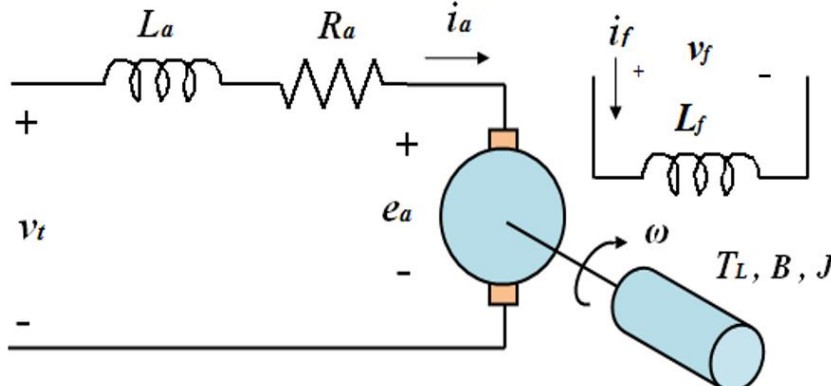

**Figure 1.** Equivalent circuit of an SEDCM.

The SEDCM model contains electrical and mechanical equations, which can be expressed as follows [4,13,15,18]:

$$v_t = e_a + L_a \frac{di_a}{dt} + R_a i_a \tag{1}$$

$$e_a = K_m \omega \tag{2}$$

$$K_m = L_{af} i_f \tag{3}$$

$$v_f = R_f i_f + L_f \left( \frac{di_f}{dt} \right) \tag{4}$$

$$T_e = K_m i_a = J \frac{d\omega}{dt} + B\omega + T_L \tag{5}$$

where $v_f$ and $v_t$ are the voltages to the field and armature, respectively; The induced emf in the armature winding is denoted by $e_a$ (volt); are the field and armature currents are denoted by $i_a$ and $i_f$, respectively; $L_f$, $L_a$, and $L_{af}$ are the armature, field, and mutual inductances, respectively; $R_a$ and $R_f$ are the armature and field resistances, respectively; $K_m$ is the constant of the motor; $T_e$ and $T_L$ are internal and load torques, respectively; J is the rotational inertia (kg/m$^2$); $\omega$ is the motor speed (rad/s); and the viscous friction of motor is denoted by B (Nm/rad/s).

### 2.2. Speed Control Method

The armature voltage control method may be used to regulate the speed of a SEDCM, which allows the speed to be varied from zero to the rated at constant torque according to the speed–torque characteristics. The speed can exceed the rated speed if the magnetic field flux is decreased [14]. In this section, only the armature voltage control has been utilized. Figure 2 depicts the general block of the SEDCM drive system [14,15].

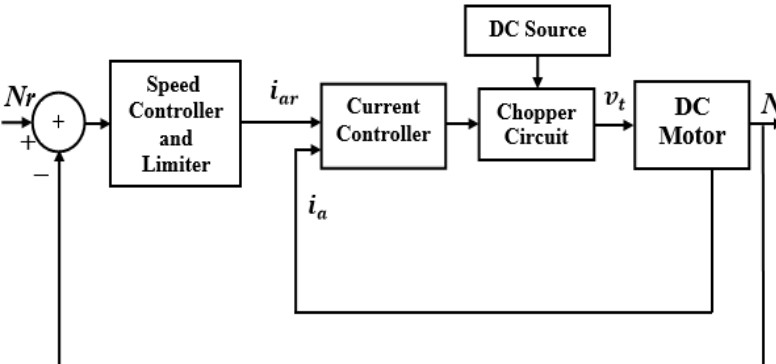

**Figure 2.** Typical block diagram of the SEDCM drive system.

This approach consists of two control loops: an outer loop representing the speed controller and an inner loop representing the current controller. A controller is utilized to obtain the control signal by comparing the motor speed to the intended speed to establish the motor's desired reference armature current. The armature current sought by the motor will influence any change in motor speed. Hysteresis current controller (HCC) is utilized in the inner loop to control the current by comparing the motor current with the desired current and generating the switching control signal for the DC–DC converter (chopper). The armature current in an HCC is forced to stay within the hysteresis band defined by the upper and lower hysteresis limits [14]. The achievement of this method is shown in Figure 3. The pulse width modulation controller depends on the chopping frequency, which can be variable or constant [18].

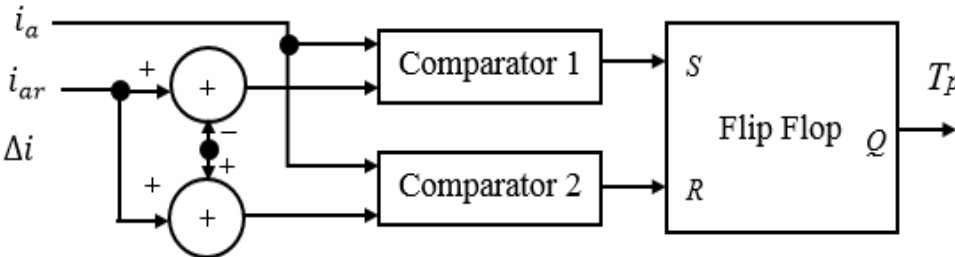

**Figure 3.** Realization of the hysteresis controller.

### 3. Fuzzy-Neural Control Scheme

The fuzzy inference system (FIS) has been extensively utilized in recent years due to its excellent performance, particularly when the system is complex, and the traditional technique cannot effectively work. Furthermore, the fuzzy control system systematically evolves human knowledge and integrates it into engineering systems. However, there is a challenge with a fuzzy control system, which is the procedure of tuning its parameters and the time it takes, which is dependent on human understanding through trial and error. Accordingly, there has been great attention recently in combining the FIS and the neural network system [20,21]. The major aim of this incorporation is to bring the features of fuzzy control systems and neural networks together to resolve the individual problems of each type before the integration between them. The combination system will possess the benefits of both systems, namely, human-like if–then rule thinking and optimization and learning abilities [22].

*3.1. Architecture of FNN*

The structural design of the four-layer FNN is depicted in Figure 4. Every node and layer have its real significance due to the configuration of the FNN, which depends on the fuzzy inference [23–25]. Every layer in Figure 4 can be defined as follows:

**First layer—input layer**: This layer is responsible for transferring the input linguistic variables $x_n$ to the output without change.

**Second layer—membership layer:** the membership layer symbolizes the input values with the Gaussian membership function as follows:

$$\mu_j^i = \exp\left(-\frac{1}{2} \frac{(x_j - c_{ij})2}{s_{ij}^2}\right) \tag{6}$$

The mean and standard deviation of the Gaussian function are denoted by $c_{ij}$ and $s_{ij}$ (i = 1, 2, . . . , n; j = 1, 2, . . . , m) in the *j*th term of the *i*th input linguistic variable $x_j$ to the node of this layer, respectively.

**Third layer—rule layer:** the rule layer executes the mechanism of the fuzzy inference. Every node in this layer multiplies the incoming signals and outputs the product. This layer's output may be expressed as follows:

$$h_i = \prod_j^n \mu_j^i \tag{7}$$

where $h_i$ represents the *i*th, the output of the rule layer.

**Fourth layer—output layer:** This layer's nodes reflect the output linguistic variables. Each node $y_o$ (o = 1, ... , $N_o$) computes the following output:

$$y_o = \sum_i^m w_i^o h_i \tag{8}$$

where $w_i^o$ represents the *i*th the rule layer 's output weight.

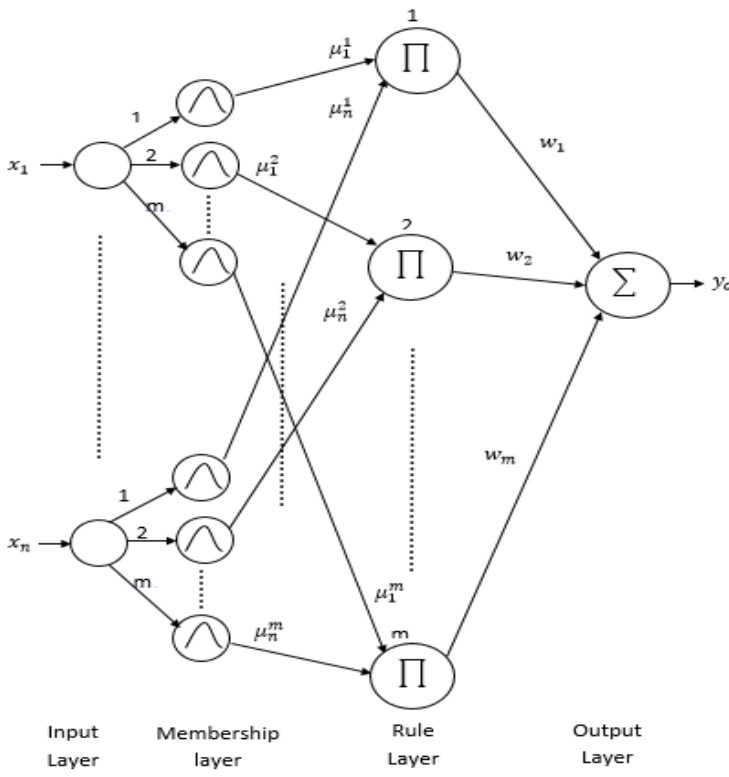

**Figure 4.** Structure of the four-layer FNN.

*3.2. Learning Algorithm for FNN*

The FNN has three types of parameters that can be adapted: the prime part—the centre values $c_{ij}$ and width values $s_{ij}$ of the Gaussian functions; and the consequence part—the output weight values $w_i$. The gradient descent algorithm provides the equations for learning the parameters [25,26]:

$$w_i(t+1) = w_i(t) - \eta_w \frac{\partial E}{\partial w_i} \tag{9}$$

$$c_{ij}(t+1) = c_{ij}(t) - \eta_c \frac{\partial E}{\partial c_{ij}} \tag{10}$$

$$s_{ij}(t+1) = s_{ij}(t) - \eta_s \frac{\partial E}{\partial s_{ij}} \tag{11}$$

the least mean square error is denoted by E, $\eta$ is the learning rate for all parameters in this system, i = 1, 2,3, ... , n and j = 1, 2, 3, ... , m.

## 4. Proposed FNMR Control Method

### 4.1. Model Reference Control

MR control was primarily suggested to resolve an issue that has provided the condition in relations of an orientation model that reports exactly how the development output must typically react to the reference signal. Figure 5 depicts a model reference adaptive control (MRAC) block diagram. In this situation, the reference model is connected in parallel with the plant. The regulator is held on two loops. The regulator and the plant form the first inner loop, which is the normal feedback. The second adaptation outer loop tunes the regulator settings to guarantee that the error e between the process output y and the model output $y_m$ becomes acceptable [27].

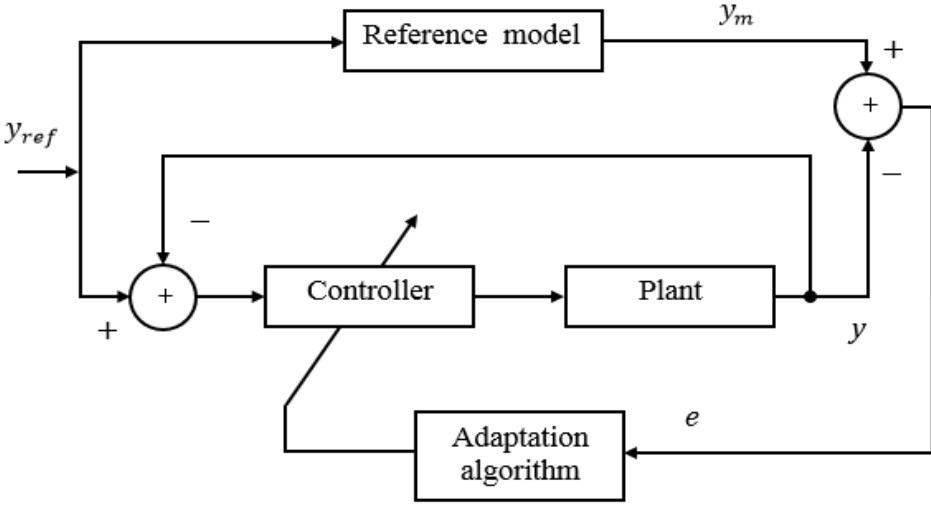

**Figure 5.** Block diagram of MRAC.

### 4.2. Mathematical Analysis of FNMRC

In nonlinear systems, the control parameters should be tuned and adapted to obtain excellent performance [19]. Several algorithms are applied using a reference model to achieve good performance for the closed-loop controller [27,28]. The controller is designed in such a way that the measured output of the plant corresponds to the desired output of the reference model. Figure 6 displays a block schematic of the SEDCM speed control utilizing the proposed FNMR with online tuning. Accordingly, The suggested controller's parameters are tuned using a gradient descent-based back-propagation technique.

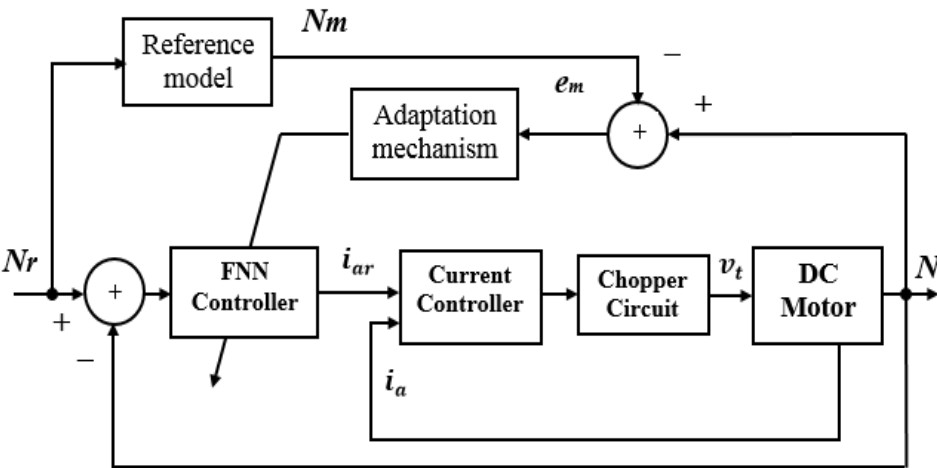

**Figure 6.** Block diagram of the proposed FNMRC control.

The reference model is characterized as a typical second-order term [20,29,30]:

$$G_{mod}(s) = \frac{\omega_n^2}{s2 + 2\zeta\omega_n + \omega_n^2} \tag{12}$$

where $\zeta$ is a damping ratio, and $\omega_n$ is a resonant frequency.

The FNMR controller parameters have been adjusted based on Equations (9)–(11). The learning procedure's goal is to reduce the difference between the reference model's actual and desired outputs. The least mean square function is used in this study to suggest a standard for error E.

$$E(t) = \frac{1}{2} e^2 \tag{13}$$

$$e = N(t) - N_m(t) \tag{14}$$

where $N(k)$ is the motor model output, $N_m(k)$ is the reference output, while the error is denoted by e. The new values of $w_i$, $c_{ij}$, and $s_{ij}$ after adaptation are equal to:

$$w_i(t+1) = w_i(t) - \eta_w \, e \, h_i \tag{15}$$

$$c_{ij}(t+1) = c_{ij}(t) - \eta_c \, e \, h_i \, w_i \, \frac{(x_j - c_{ij})}{s_{ij}^2} \tag{16}$$

$$s_{ij}(t+1) = s_{ij}(t) - \eta_s \, e \, h_i \, w_i \, \frac{(x_j - c_{ij})^2}{s_{ij}^3} \tag{17}$$

*4.3. Multi-Motor System*

A multi-motor system is the linkage of different motors in such a way that their synchronization is maintained. However, in the traditional master–slave method, the first motor is selected as the master motor that achieves the chosen trajectories, while the slave motor tracks the motion of the master motor [31–34]. Figure 7 describes the structure of the master-slave. The performance of path tracking can be reasonably limited because of the certainty that the real trajectory of the master motor acts as the controlled path of the slave motor.

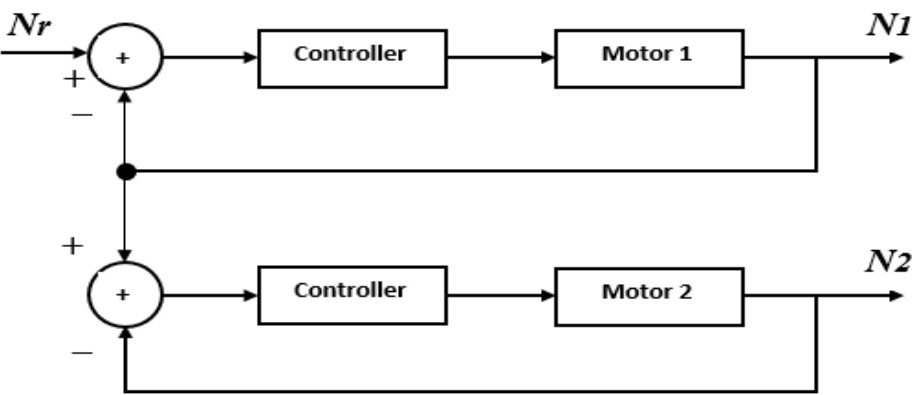

**Figure 7.** Block diagram of the master–slave system.

However, the two motors in the system receive the same set point are designed in the set point coordinated control method. This notion means that the slave motor does not become dependent on the master motor. Figure 8 depicts the block diagram of the set-point coordinated regulator structure. The main issue with this method of control is that the load disturbances of the motors or their dynamic parameter change. Given that FNNs have the ability to avoid changes in loads and high efficiency, using them in the multiple-motor

system will contribute to the elimination of all disadvantages in the above-mentioned methods [31,35,36].

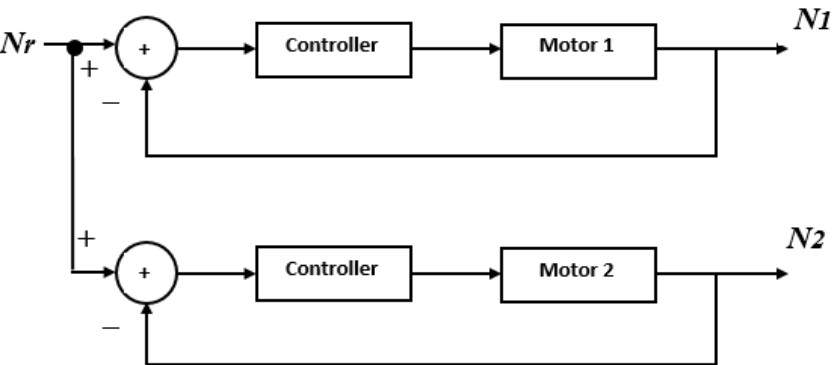

**Figure 8.** Block diagram of the set point coordinated regulator system.

## 5. Simulation Results and Discussion

The proposed models in Matlab/Simulink for the FNMRC controller of the SEDCM and its model are depicted in Figure 9. The proposed program is written in Matlab software using the S-function tool to simulate the FNMR controller, as depict in Figure 10. The inputs to the controller have been represented by the error and its change. The error $e_m$ between the output and the model reference of this motor is utilized as a path in the adaptive mechanism. Every input of this controller contains five membership functions. Accordingly, the number of weights in the outcome part is also five. In addition, each parameter has the learning rates of $\eta_w = 0.6$, $\eta_c = 0.4$, and $\eta_s = 0.2$. The parameters of the motors are listed in Table 1. Good parameter values have been obtained from the reference model of $\zeta = 0.97$ and $\omega_n = 10$ by utilizing the trial-and-error method.

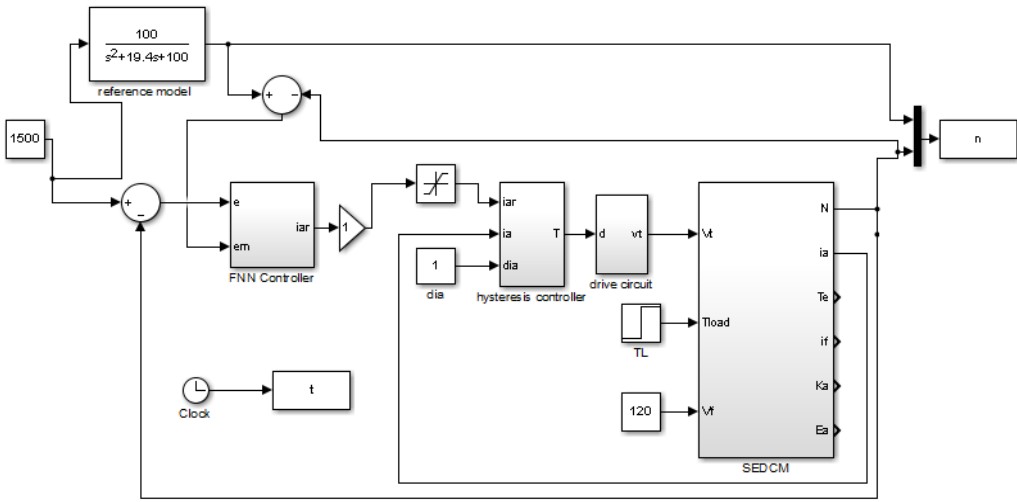

**Figure 9.** The proposed FNMR control for the SEDCM in Matlab/Simulink.

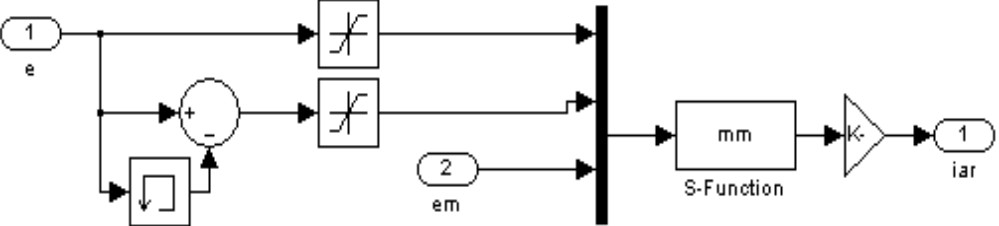

**Figure 10.** Simulink model of the FNMR controller.

**Table 1.** Parameters of the DC motors.

| Parameter | Value |
|---|---|
| N | 1750 rpm |
| $v_a$ | 230 V |
| $v_f$ | 120 V |
| $i_a$ | 46 A |
| $i_f$ | 1.6 A |
| $L_a$ | 0.767 H |
| $L_f$ | 0.008 H |
| $R_a$ | 0.1 Ω |
| $R_f$ | 75 Ω |
| B | 0.314 Nm·S/rad |
| J | 2.2 kg/m$^2$ |

The proposed system has been tested at 10 and 100 epochs. The result presents that the motor speed follows the reference speed by increasing the number of epochs. Accordingly, the error is near zero. If the reference speed input has a sinusoidal shape, then the rotor speed response at 10 epochs is well tracked by the reference model, as demonstrated in Figure 11. Figure 12 shows the rotor speed at 100 epochs. Therefore, the rotor speeds are matched together, and this is the goal of the proposed controller.

The square function is used to test the system performance. The system responses for 10 and 100 epochs are depict in Figures 13 and 14, respectively. Figure 15 reports the error values between the model reference and the speed of the motor at 100 epochs. A small error between the speeds is obtained, and this explains the aim of the proposed controller. Figure 16 reports the speed of the motor at 1500 rpm for the load torque of 25 N·m at $t = 2$ s. Figure 17 show the relationship between the cost function and epochs.

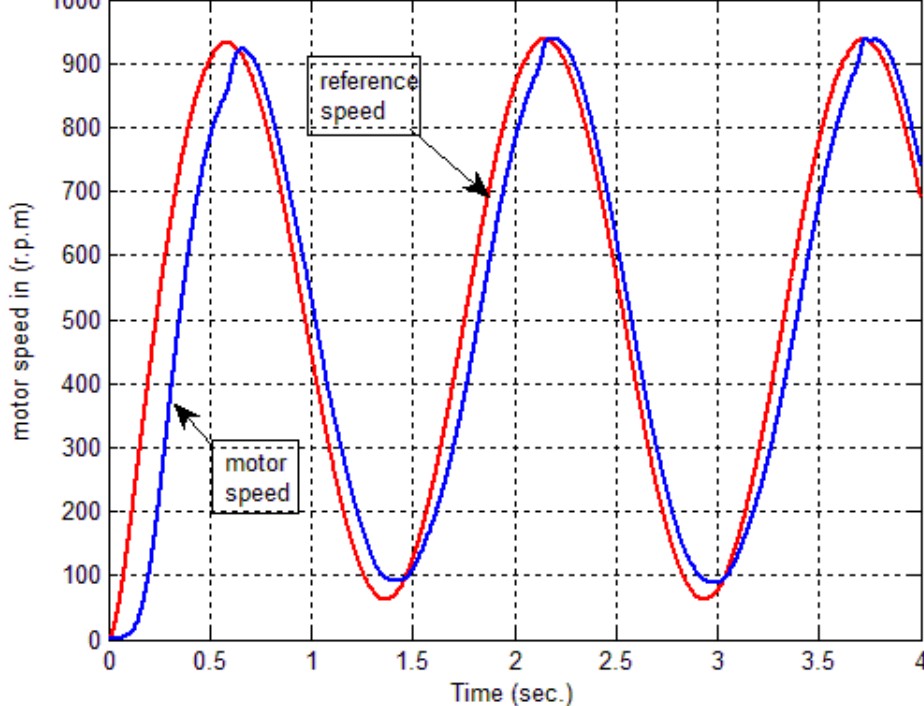

**Figure 11.** Motor response tracking reference model at 10 epochs for the sinusoidal shape of reference speed.

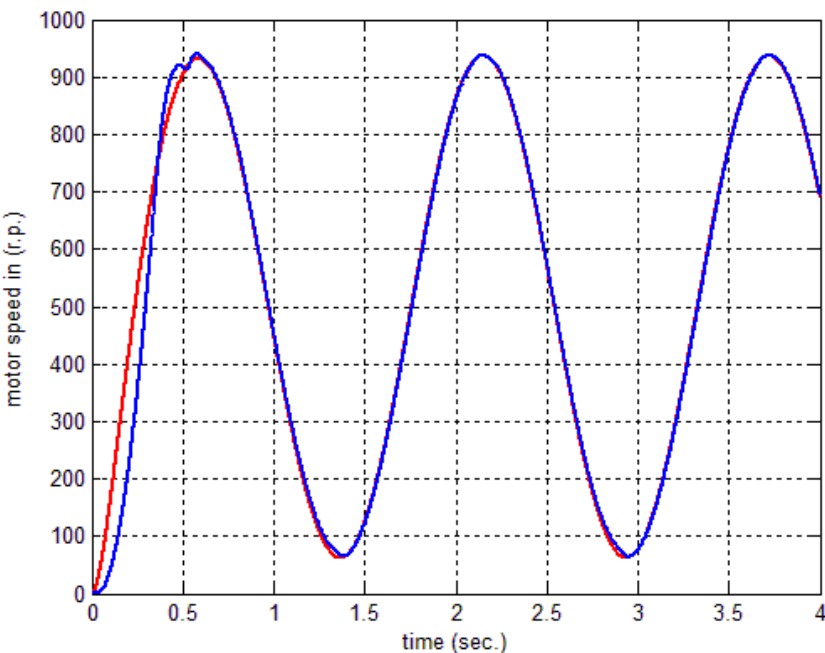

**Figure 12.** The response of the motor at 100 epochs for the sinusoidal shape of reference speed.

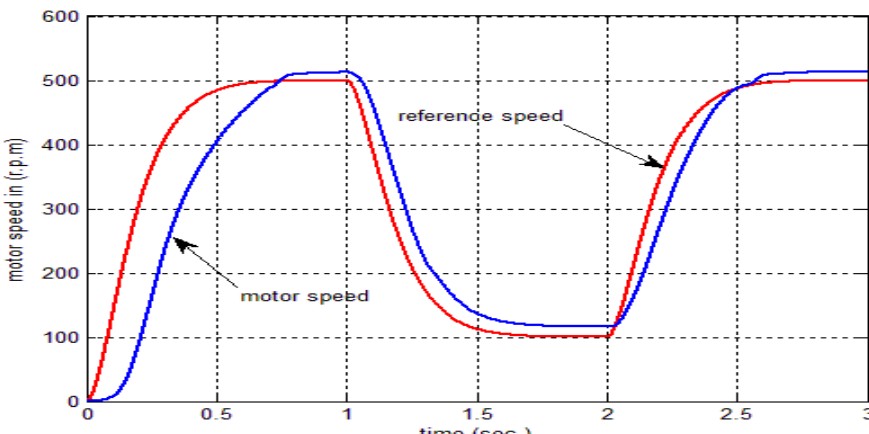

**Figure 13.** Motor response tracking reference model at 10 epochs for the square shape of reference speed.

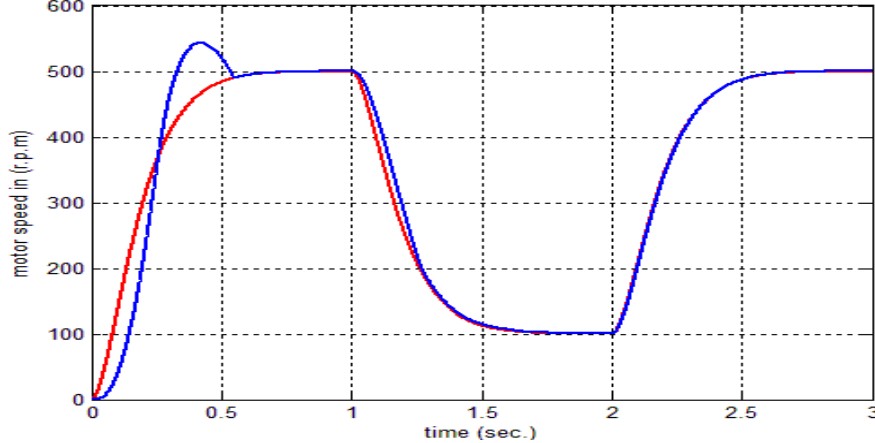

**Figure 14.** The response of the motor at 100 epochs for the square shape of reference speed.

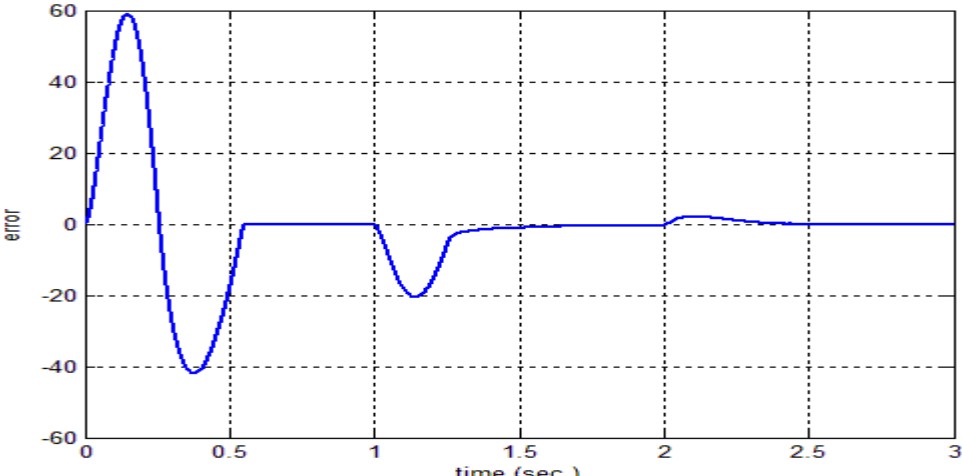

**Figure 15.** Error between the motor speed and the model reference at 100 epochs.

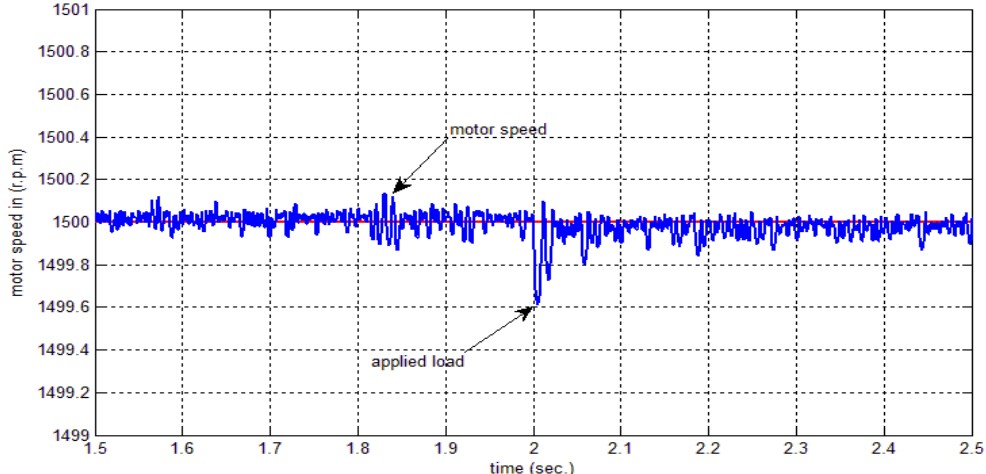

**Figure 16.** Motor speed with a step-change in the load torque.

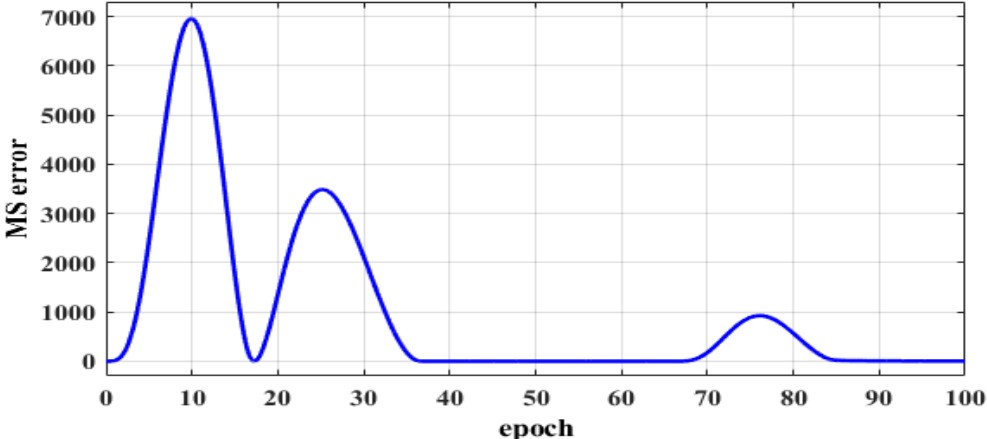

**Figure 17.** The relationship between the cost function and epochs.

Figure 18 depicts the armature current for the applied load at a time of 2 s. The response of the two motors connected in the master–slave scheme when applying an FNMRC controller is depicts in Figure 19. As seen in this figure, the slave has been synchronized with the master according to the reference trajectory. Figure 20 shows the error between the master motor speed and the slave motor speed when applying an

FNMRC controller. The error is considerably small, indicating the synchronization between the two speeds. When the load disturbances (25 N·m) at 1400 rpm speed is applied to the motors at 2 s, the sequential and synchronization of the two motors is not affected by this disturbance, as seen in Figure 21.

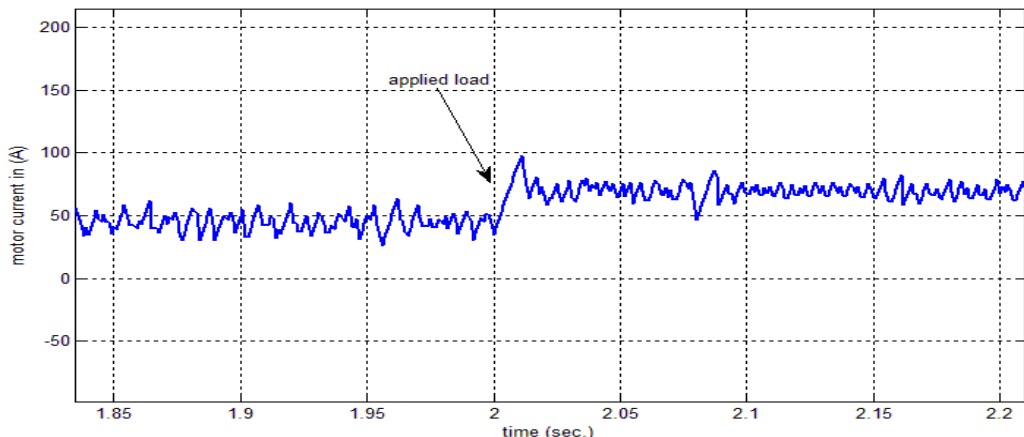

**Figure 18.** Armature current and motor speed at the applied load.

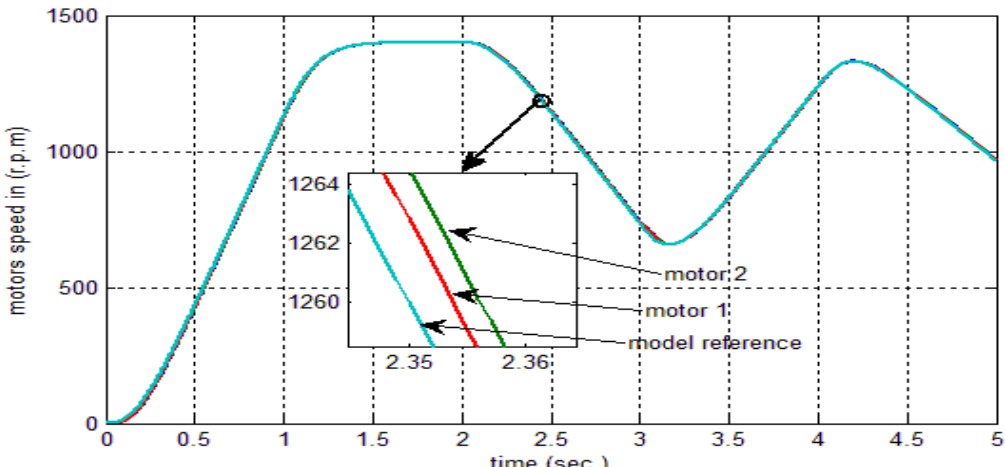

**Figure 19.** Response of the two motors connected using proposed FNMRC controller-based master–slave method.

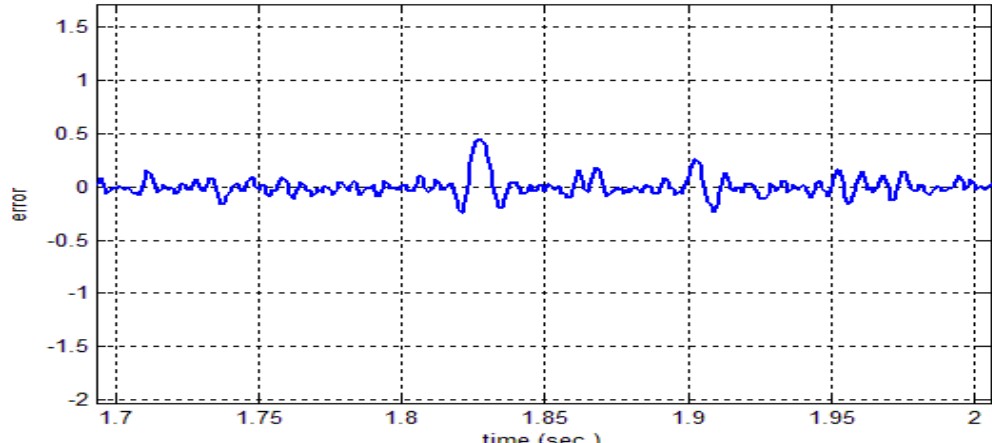

**Figure 20.** Error between the master and the slave motor speed using FNMRC controller.

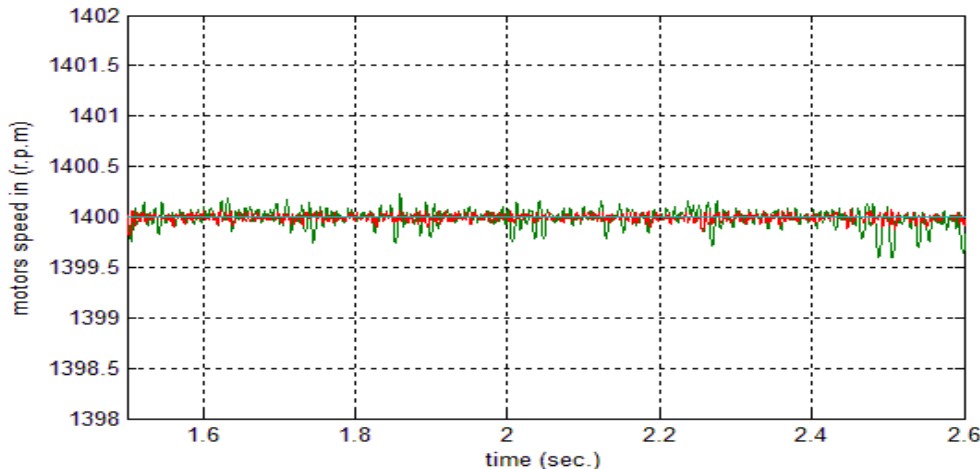

**Figure 21.** Synchronization of the two motors at load disturbance.

The response of the two motors connected in the master–slave scheme when applying a PI controller is depicts in Figure 22. Figure 23 depits the speed error between the master and the slave motor during the change of reference speed when applying a PI controller-based master–slave method. This error indicates no synchronization in speed between these motors in some regions of the operation.

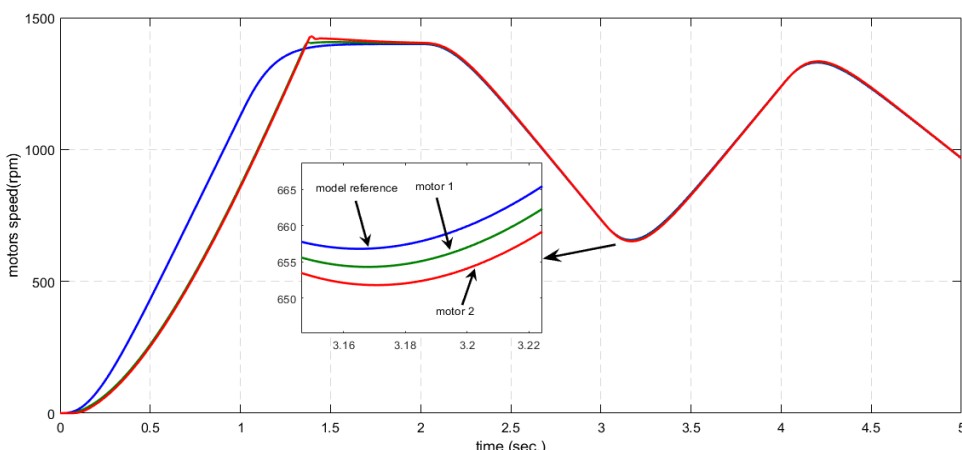

**Figure 22.** Response of the two motors connected using PI controller-based master–slave method.

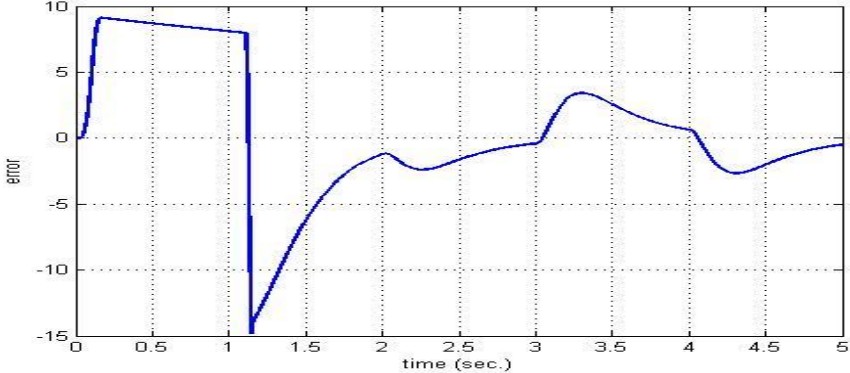

**Figure 23.** Speed error between the master and the slave motor during the change of reference speed using PI controller-based master–slave method.

## 6. Conclusions

This work presents the FNMR to control and stabilize the speed of the multi-DC motor system. The presented control has been used in the online training to control the speed of SEDCMs depending on the fitness function. The fitness function is the error between the actual speed motor and the model desired output to learn the internal parameters of the FNN, where the error can be minimized to zero. The master–slave scheme has been used to structure interconnections of two motors and keep the synchronization between them. The results depict that the proposed controller has been improved the speed response regardless of the load disturbances and accomplished good performance based on a flexible and adapted approach. The most important advantage of the proposed method is the ability to learn and adapt to the internal and external disturbances of any system to be controlled. There is also the possibility to implement this method in real time with the presence of a basic limitation, which is that this method needs a microprocessor with a suitable speed for the purpose of conducting the adaptation process online. It can be said here that the current development in the performance of microcontrollers can easily implement this method in real time.

The proposed FNMR controller can be used in multi-AC motors such as hot rolling mill and cold rolling mill systems.

**Author Contributions:** Conceptualization, A.L.S. and W.I.B.; methodology, W.I.B., A.L.S. and K.A.M.; software, W.I.B., A.L.S. and K.A.M.; validation and formal analysis, S.J.Y., M.A.Q., N.T.A. and A.N.; investigation, S.J.Y., M.A.Q. and A.N.; resources, W.I.B., A.L.S. and J.F.A.-A.; data curation, A.L.S., N.T.A., W.I.B. and S.J.Y.; writing-original draft preparation, W.I.B., A.L.S., A.N. and M.A.Q.; writing-review and editing A.N., A.L.S., W.I.B. and M.A.Q.; visualization, W.I.B., S.J.Y., A.N. and M.A.Q.; supervision, K.A.M., A.L.S. and S.J.Y.; project administration, K.A.M.; funding acquisition, J.F.A.-A. and M.A. funding acquisition. All authors have read and agreed to the published version of the manuscript.

**Funding:** This research was funded by Taif University, project number (TURSP-2020/211), Taif University, Taif, Saudi Arabia.

**Institutional Review Board Statement:** The study did not involve humans.

**Informed Consent Statement:** The study did not involve humans.

**Data Availability Statement:** Not applicable.

**Acknowledgments:** The authors appreciate Taif University Researchers supporting project number (TURSP-2020/211), Taif University, Taif, Saudi Arabia.

**Conflicts of Interest:** The authors declare no conflict of interest.

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
