# Peer review of "Speed Control of a Multi-Motor System Based on Fuzzy Neural Model Reference Method"

_actuators, doi:10.3390/act11050123_

Round 1

Reviewer 1 Report

Manuscript title: Speed Control of a Multi-motor System Based on Fuzzy Neural Model Reference Method.

The manuscript is well organized and the contents fit with the journal’s topics. The methodology is well described and applied.

However, it also presents several flaws that need to be addressed before considering it for publication.

- Introduction - What are your contributions?

- Why did you use proposed neuro-fuzzy modle? Why not other tools in the filed, like NN and meta-heuristics? You should discuss this.

- Better highlight novelty in the study.

- Better define motivations for the research.

-The analysis of the literature is a very weak point of this paper and must be improved. In the entire paper, there is only three cited paper from 2020-2021. Add another papers (10-15) of more recent date (period 2020-2022), such as: Comparison of robust optimal QFT controller with TFC and MFC controller in a multi-input multi-output system, https://doi.org/10.31181/rme200101151g; Modeling of neuro-fuzzy system as a support in decision-making processes, https://doi.org/10.31181/rme2001021222b. Good academic practice implies that the main attention in the analysis of literature is directed towards new research. These are usually studies that have been published in the previous two, possibly three years.

-Based on literature analysis author should better highlight the objective of their paper and to what extent it contributes to close a gap in the existing literature and/or practice. What is the innovative value of the contribution proposed by the authors? This is an essential part of the Introduction section.

- In conclusion: Show in detail the advantages and limitations of the proposed methodology and this study; Add future research.

Author Response

Responses to the Editor’s and Reviewers' Comments

We appreciate very much the editor and the reviewers for the constructive comments. We also thank the editor and the reviewers for the effort and time put into the review of the manuscript. Each comment has been carefully considered point by point and responded. Responses to the reviewers and changes in the revised manuscript are done.

Reviewer 2 Report

General

It is quite interesting paper reporting the Authors’ efforts towards applying of the DC motor speed control system based on model reference approach with the use of fuzzy neural networks. However, in fact, the control idea based on minimizing difference of the model reference and plant output should not be recognized as novel. It is well known since many years.

The significant part of the paper is devoted to the proposition of an adaptive FNN controller. The FNN structure and its learning procedure is classic and well known. In this scope the paper also does not bring any significant contribution. 

The motivation for introducing FNN are nonlinearities of the controlled system.  However, they are poorly explained and exposed in the paper. For example, there is nothing said or shown how behaves the control system in case of variation of Coulomb friction. It is suggested to extend appropriately the paper.

It will be also profitable to show how the different loads affect control error. The test based on only one change of load is not enough. Please consider that angular velocity measurement used in simulation does not reflect limited resolution and delay of real velocity measurements. It may have noteworthy influence on tracking error.

To make the paper more readable please consider explaining how the parameters of the reference model are identified (see Eq.12 and Fig. 9).

I also encourage Authors to give some examples of the foreseen applications of the proposed control approach.

Novelty and contribution

I found the contribution of the paper in a promising approach to control system with the use of model- based control concept.  The exemplary results of simulations show some potential in the proposed approach. However, nothing could be said if simulation results will not be confirmed by results of experimental investigations. The experience shows that simulation and real worlds are quite different in most cases.

What is missing?

  1. Experimental verification of simulation results.
  2. Identification of parameters of reference model.
  3. Explanation how was tuned PI controller.
  4. Foreseen applications.
  5. Stability analysis or discussion.

Detail

  1. Rows 43-44.

Please check the sentence. It seems not to be completed.

  1. Row 75.

Typo.

  1. Row 76.

Please explain MRAC abbreviation.

  1. Row 82.

Please explain SEDCM abbreviation.

  1. Rows 186-188.

Please explain symbol E.

  1. Row 257

Please explain what exactly means high control efficiency.

  1. Figures referring to control error.

For easier comparison please consider showing normalized control error.

Author Response

(The authors gave the same response as above.)

Reviewer 3 Report

This work presents the FNMR to control and stabilize the speed of the multi-DC motor system.

In general, authors present the FNMR to control and stabilize the speed of the multi-DC motor system. Authors should consider the following comments to clarify the main contributions of their paper.  

1.- In the page 1, in the introduction, authors say “These controllers have certain drawbacks, such as load disturbance and sensitivity to a variety of motor parameters.”, in this text they should include references [a]-[f] which also consider controllers with disturbances for motors.

[a] On the Rejection of Random Perturbations and the Tracking of Random References in a Quadrotor, Complexity, Vol. 2022, 1-16, 2022.

[b] Modified Linear Technique for the Controllability and Observability of Robotic Arms, IEEE Access, Vol. 10, 3366-3377, 2022.

[c] PI-Type Controllers and Σ–Δ Modulation for Saturated DC-DC Buck Power Converters, IEEE Access, Vol. 9, pp. 20346-20357, 2021.

[d] Optimization of Sliding Mode Control to Save Energy in a SCARA Robot, Mathematics, Vol. 9, No. 24, 3160, 2021.

[e] PD Control Compensation Based on a Cascade Neural Network Applied to a Robot Manipulator, Frontiers in Neurorobotics, Vol. 14, 2020.

[f] Sensorless tracking control for a full-bridge Buck inverter-DC motor system: Passivity and flatness-based design, IEEE Access, Vol. 9, pp. 132191-132204, 2021.

2.- In the page 4, authors say “In this chapter, only the armature voltage control has been utilized.”, it should be “In this section, only the armature voltage control has been utilized.”

3.- In the page 5, in the equation (6), authors should clarify if there is a special reason to consider the Gaussian membership function.

4.- In the page 6, in the equations (9)-(11), authors should clarify if there is a special reason to consider the gradient descent algorithm.

5.- In the page 9, in the simulation results, authors should clarify if they compare their method with other previous.

6.- In the page 16, in the conclusion, authors should include some future research.

Author Response

(The authors gave the same response as above.)

Round 2

Reviewer 1 Report

All the reviewers' comments have been addressed carefully and sufficiently. The revisions are rational from my point of view. I think the current version of the paper can be accepted.